# The Procaine-Based ProcCluster^®^ Impedes the Second Envelopment Process of Herpes Simplex Virus Type 1

**DOI:** 10.3390/ijms26157185

**Published:** 2025-07-25

**Authors:** Johannes Jungwirth, Lisa Siegert, Lena Gauthier, Andreas Henke, Oliver H. Krämer, Beatrice Engert, Christina Ehrhardt

**Affiliations:** 1Section of Experimental Virology, Institute of Medical Microbiology, Center for Molecular Biomedicine (CMB), Jena University Hospital, Hans-Knoell-Str. 2, D-07745 Jena, Germany; johannes.jungwirth@med.uni-jena.de (J.J.); lena.gauthier@leibniz-ipht.de (L.G.); andreas.henke@med.uni-jena.de (A.H.); 2Institute of Toxicology, University Medical Center of the Johannes Gutenberg University Mainz, Obere Zahlbacher Str. 67, D-55131 Mainz, Germany; okraemer@uni-mainz.de; 3Department Biophysical Imaging, Leibniz Institute of Photonic Technology (IPHT), Member of the Leibniz Center for Photonics in Infection Research (LPI), Albert-Einstein-Str. 9, D-07745 Jena, Germany; 4Institute of Applied Optics and Biophysics, Friedrich Schiller University Jena, Max-Wien-Platz 1, D-07743 Jena, Germany; 5inflamed pharma GmbH, Winzerlaer Str. 2, D-07745 Jena, Germany; b.engert@inflamedpharma.com

**Keywords:** HSV-1, antiviral, host cell-targeting, endosomes, Rab GTPases, drug repurposing, ProcCluster^®^

## Abstract

Herpes simplex virus type 1 (HSV-1) has a global prevalence of 64%. Established antiviral drugs, such as acyclovir (ACV), have been successfully used over the past decades. However, due to growing viral resistance against approved antivirals and the lack of effective vaccines, new concepts are essential to target HSV-1 infections. Here, we present data on the inhibitory effect of the procaine-based substance ProcCluster^®^ (PC) in reducing HSV-1 replication in vitro. Non-toxic PC concentrations significantly decreased HSV-1 replication in infected cells. Immunofluorescence microscopy revealed an accumulation of viral proteins in early and recycling endosomes, resulting in reduced viral release. The combination of PC with ACV resulted in an enhanced antiviral effect. Based on these results, PC alone, as well as in combination with ACV, appears to be a promising substance with antiviral potential against HSV-1 infections.

## 1. Introduction

The herpes simplex virus type 1 (HSV-1) contains a double-stranded DNA (dsDNA) genome and belongs, together with HSV-2 and the varicella-zoster virus, to the subfamily of the α-*herpesvirinae* [1]. The World Health Organization (WHO) reports that globally 3.8 billion people (64.2%) under the age of 50 are infected with HSV-1 [2]. In humans, HSV-1 primarily targets epithelial cells of the skin or the mucosa but can also infect the eye, causing corneal inflammation [3]. Contact with virus-containing mucus and other body fluids may evoke clinical manifestations characterized by cold sores at the orolabial region and/or a corneal keratitis [4]. Severe HSV-1 infections result in encephalitis, meningitis, or blindness in newborns or immunocompromised individuals and are often associated with a systemic HSV-1 distribution [5].

The initial infection is followed by HSV-1 replication at the entry site and subsequent transport of the virus through axons of sensory nerves to sensory ganglia [4]. By establishing latency in the nervous system, HSV-1 evades the host’s immune response and reduces its exposure to antiviral drugs [6]. During latency, viral genomes persist in the nucleus of neurons without producing new viral proteins [7]. Periodically, viral replication can be reactivated by various factors such as stress, fever, ultraviolet light, or diseases. In such cases, newly assembled viral particles travel from sensory nerves back to the primary area of infection causing local inflammation [7].

HSV-1 consists of three distinct structures: nucleocapsid, tegument, and envelope. The nucleocapsid protects the linear dsDNA genome of approximately 152 kbp in the inner core [8]. The capsid is surrounded by the tegument, a protein-rich layer composed of over 20 different virus-encoded proteins (VPs), providing a connection between the capsid and the envelope [9]. The viral particle is enclosed by an envelope, which contains 16 different VPs, including 12 glycoproteins for regulating viral attachment, entry, and cell-to-cell spread [9,10].

The entry of HSV-1 depends on the interaction of viral glycoproteins with host cell factors that differ between tissues and cell types. The virus can enter a cell via direct fusion with entry receptors at the plasma membrane or via endocytosis [11]. After membrane fusion, an interaction of tegument proteins with cellular motor proteins, such as dynein or kinesin, facilitates the transport of capsids along microtubules to the nucleus [12]. The viral DNA enters the nucleus through nuclear pores and viral gene expression is initiated in a cascade-dependent manner [13]. Translated viral immediate-early (IE) proteins promote the transcription of early (E) genes, which themselves are involved in viral DNA replication and transcription of late (L) genes [14]. The nucleus is the site of viral genome replication and assembly of new capsids. DNA-containing capsids bud at the inner nuclear membrane to form an enveloped particle in the perinuclear space. This envelope is lost after fusion with the outer nuclear membrane, releasing capsids into the cytosol [9,15,16]. Viral glycoproteins are transcribed and processed via the conventional secretory pathway. They traffic from the endoplasmic reticulum (ER) to the trans-Golgi network, followed by a Rab GTPases (Ras-related protein in brain guanosine triphosphatases)-dependent transport to the plasma membrane [17,18,19]. Thereafter, viral glycoproteins on the cellular surface are endocytosed into vesicular structures that provide the envelope for the capsids [17,18,19]. This second envelopment in the cytoplasm depends on the viral glycoprotein gD. During replication of HSV-1 strains without functional glycoprotein gD, numerous non-enveloped capsids accumulate in the cytoplasm of host cells along with partially enveloped capsids and morphologically aberrant enveloped particles [20]. Enveloped virions are exocytosed via membrane vesicles that fuse with the plasma membrane, resulting in the egress of HSV-1.

The endocytic pathway is regulated at different stages by a number of Rab GTPases [21]. These proteins cycle between their inactive guanosine diphosphate (GDP)-bound form and their active guanosine triphosphate (GTP)-bound form. Active forms coordinate cellular cargo movement and, in infected cells, viral protein trafficking [22]. More than 70 different Rab GTPases are not only involved in trafficking, but also in other cellular events, including autophagy and cell cycle regulation. Dysfunctional Rab GTPases are involved in the development of genetic diseases or cancer [23,24,25].

Different viruses infecting eukaryotic cells rely on pathways associated with Rab GTPases [26]. During these processes, viral entry is frequently mediated by Rab5 and Rab7, which regulate endosomal trafficking, whereas Rab6 and Rab9 are involved in intracellular transport pathways. Rab11 is essential for viral egress [26]. Epstein–Barr virus (EBV), a member of the γ-*herpesvirinae*, is released to the extracellular milieu via a secretory pathway employing Rab8A, Rab10, and Rab11. Knockdown of these proteins causes intracellular accumulation of EBV proteins and prevents the release of infectious virions [27]. In the case of human immunodeficiency virus (HIV), Rab7A regulates envelope processing, incorporation of envelope proteins into new viral particles, and infectivity of the virus [28]. Moreover, Rab11 is involved in viral ribonucleoprotein (vRNP) trafficking of the influenza A virus (IAV) [29,30,31,32], and in budding events of the respiratory syncytial virus (RSV) [33]. Moreover, hepatitis B virus (HBV) replication depends on Rab5A and Rab7A activity for viral uptake and transport to the nucleus. Furthermore, Rab33B and Rab27 are involved in the transport of HBV through the Golgi apparatus, as well as exosomal release [34].

While HSV-1 infections are treatable, they are not curable. So far, antiviral drugs primarily target viral DNA replication, reducing healing time and symptom severity [35]. Today, antiviral compounds against HSV-1 can be divided into four groups: nucleoside analogues, nucleotide analogues, helicase-primase inhibitors, and pyrophosphate analogues [36,37]. Acyclovir (ACV), an antiviral drug approved already in 1977, is a synthetic acyclic nucleoside analogue of guanosine. After activation by the viral thymidine kinase (TK), it competes with the natural nucleotide dGTP and disrupts the function of the viral DNA polymerase (POL), decreasing viral replication [38]. Structurally related drugs to ACV, like penciclovir, as well as their prodrugs (e.g., valaciclovir, valganciclovir, and famciclovir), offer better bioavailability and higher selectivity [39,40,41]. Nucleotide analogues like cidofovir (CDV) and brincidofovir (CDV, conjugated with a lipid molecule) are analogues of cytosine monophosphate and do not require phosphorylation by viral TK to inhibit viral POL activity [36]. Helicase-primase inhibitors, such as amenamevir, prevent the separation of viral DNA strands and do not require virus-induced phosphorylation to generate the active form [42,43]. Non-nucleoside analogues of pyrophosphate, like foscarnet, directly inhibit the viral POL by binding to the pyrophosphate binding site, and blocking the cleavage of nucleoside triphosphates [36,40]. Hence, foscarnet does not require activation by viral TK. Therefore, it is used to treat infections with ACV-resistant HSV-1 strains [44]. The risk of developing viral drug resistance due to TK or POL polymorphisms increases in patients who require prolonged antiviral treatment [45,46]. For this reason, repurposing existing drugs that have already proven to be safe in the treatment of other diseases, and/or the implementation of multidrug therapy, represents a promising alternative [47,48]. The synergistic combinations of two or more substances with antiviral potential suggest a treatment advantage and are already established in clinical routine [49,50,51,52,53].

The local anesthetic procaine was first synthesized in 1904. Next to its narcotic effect, it shows antiviral potential against several viruses, including HSV-1 [54,55]. Treatment with other anesthetics, like lidocaine, dibucaine, or tetracaine, reduced the infectivity of HSV-1 directly [56]. A topical administration of an anesthetic cream containing lidocaine and prilocaine showed an efficient decrease in virus-caused lesions in HSV-1-infected mice and humans [57,58]. ProcCluster^®^ (PC) is a substance based on sodium chloride-clustered procaine-hydrogencarbonate and has recently been shown to have antipathogenic activity against severe acute respiratory syndrome coronavirus 2 (SARS-CoV-2), influenza A virus (IAV), and/or *Aspergillus fumigatus* in vitro [59,60,61].

In the present study, we investigated the antiviral properties of PC against HSV-1 infection in vitro and recognized that it decreases HSV-1 replication at low millimolar concentrations across different cell lines and HSV-1 strains. We observed that PC is acting late in the HSV-1 replication cycle making it a potential combinable candidate with ACV.

## 2. Results

### 2.1. ProcCluster^®^ Inhibits HSV-1 Replication

HSV-1 establishes primary infections in human lips or eyes, followed by a latent state in neurons [62]. Human retinal pigmented epithelium cells (hTERT RPE-1) and human keratinocytes (HaCaT) are established host cells for HSV-1. Neuroblastoma cell lines are frequently used as models for neuronal HSV-1 infections [63,64,65]. To determine the range of non-toxic concentrations, the influence of PC on cellular metabolic activity, compared to solvent controls, was studied via MTT assays (Figure 1a). PC caused a 50% reduction in the metabolic activity of RPE-1 cells at 4.17 mM (95% confidence interval (CI) 2.93–5.98 mM), of HaCaT cells at 6.27 mM (CI 5.08–7.87 mM), and of Kelly neuroblastoma cells at 2.64 mM (CI 2.48–2.80 mM). While treatment concentrations up to 2.5 mM did not significantly affect the metabolic activity of both epithelial cell lines, a slight effect was observed in the neuroblastoma cell line. In order to maintain comparable experimental conditions, all three cell lines were treated with the same amounts of PC.

Infection experiments of RPE-1, HaCaT, and Kelly cells were used to determine whether PC treatment revealed inhibitory effects on HSV-1 replication. Cells of all three cell lines were infected with a multiplicity of infection (MOI) of 1.0 using two different HSV-1 strains: the laboratory strain HSV-1 (Figure 1b) and the clinical isolate HSV-1 1223/1999 (Figure 1c). After infection, cells were treated with 0.63 to 2.5 mM PC until 24 h post-infection (p.i.). Overall, a dose-dependent significant reduction in virus replication was observed in all experiments.

### 2.2. ProcCluster^®^ Application Induces the Accumulation of the Viral Glycoprotein gD with Endosomal Host Cell Proteins

PC treatment had no effect on the expression of the IE infected cell protein 0 (icp0), the E protein icp8, and the glycoprotein gD of HSV-1 in immunoblot analyses (Appendix A), indicating that in the presence of PC only the release of mature HSV-1 particles is reduced. This was confirmed by comparison of released and cell-associated viral particles.

Analysis of the concentration of cell-associated viral particles at 24 h p.i. in PC-treated, HSV-1-infected RPE-1 cells revealed no reduction in infectious virions compared to the untreated controls (Appendix A).

To investigate late replication events of HSV-1 during PC treatment, immunofluorescence microscopy was performed. To study the cellular localization of the viral glycoprotein gD, HSV-1-infected RPE-1 cells were treated with 2.5 mM PC. After 24 h, fixed cells were labeled with antibodies directed against the viral glycoprotein gD, the marker Rab5, specific for early endosomes, (Figure 2a–f), the marker Rab11, specific for recycling endosomes, (Figure 2g–l), the marker Rab7, specific for late endosomes (Figure 2m–r), or the marker LAMP1-1, specific for lysosomes, (Figure 2s–x). Consistent with the cell-associated total viral concentration (Appendix A), measurement of the mean fluorescence intensity of viral glycoprotein gD in PC-treated, HSV-1-infected RPE-1 cells indicated no increased intensity (Appendix A). Microscopic images of uninfected RPE-1 cells revealed a weak distribution of the endosome markers Rab5 and Rab11, regardless of whether PC was added or not (Figure 2a,b,g,h). In contrast, HSV-1 infection induced an accumulation of both Rab GTPases (Figure 2c–f,i–l). Furthermore, in H_2_O-treated cells, HSV-1 mainly localized at the plasma membrane with only partial overlap with both endosomal markers as visible in ortho-view images (Figure 2c,e,i,k). PC treatment, however, resulted in the accumulation of larger intracellular vesicles and a strong correlation and co-occurrence of viral glycoprotein gD together with Rab5 or Rab11, respectively, within these vesicles (Figure 2d,f,j,l, Figure 3 and Appendix A). Both Pearson’s correlation, as well as Manders’ thresholded coefficients show a significant increase in their respective coefficients (Figure 3 and Appendix A).

To analyze, whether that correlation and co-occurrence of viral glycoprotein gD with Rab5 or Rab11 in the presence of PC is restricted to early and recycling endosomes, respectively, the viral glycoprotein gD was additionally assayed for PC-induced co-localization with the late endosomal marker protein Rab7 (Figure 2m–r) and the lysosomal marker protein LAMP-1 (Figure 2s–x). In uninfected RPE-1 cells, both marker proteins Rab7 and LAMP-1 were evenly distributed (Figure 2m,n,s,t). However, HSV-1 infection did not notably affect Rab7 and LAMP-1 distribution (Figure 2o–r,u–x). Strikingly, PC treatment resulted again in enlarged viral glycoprotein gD clusters but this time neither an increased correlation, nor an increased co-occurrence was observed with either Rab7 or LAMP-1 (Figure 2q,r,w,x, Figure 3 and Appendix A). In parallel, both Pearson’s coefficients for Rab7 and LAMP-1 are lower, demonstrating a weaker correlation between gD and both markers. Therefore, our data might indicate that PC treatment causes viral glycoprotein gD accumulation in early and recycling endosomes, but not in late endosomes or in lysosomes.

### 2.3. HSV-1 Replication Is Strongly Inhibited by the Combinatorial Treatment with PC and ACV

Using a combination of two or more drugs represents an alternative strategy to increase the efficiency of established antiviral therapies. Therefore, the effect of a combinatorial application of PC and ACV on HSV-1-infected RPE-1 cells was investigated. Virus-infected cells were treated with PC-ACV combinations in different concentrations. For all tested combinations, a potent reduction in virus concentrations and an increased inhibition of HSV-1 replication were detected, especially in samples treated with 1 mM PC and 1 µM ACV (Figure 4).

## 3. Discussion

HSV-1 effectively establishes acute and latent infections in humans by evading host antiviral innate immunity. It employs multiple strategies to counteract immune defenses, and modulates cellular survival pathways. Several studies have shown that HSV-1 has evolved numerous measures to manipulate toll-like receptor (TLR) signaling, retinoic acid inducible gene (RIG)-I- and melanoma-differentiation-associated protein (MDA)5-mediated antiviral signaling, DNA sensor-mediated interferon (IFN)-I signaling, nucleotide-binding domain leucine-rich repeat (NLR) sensors responsible for the detection of invading pathogens, DNA damage response signaling, ER stress response resulting from the translation of viral proteins, autophagy, apoptosis, and necrosis [66]. Developing host-directed therapies against HSV-1 infections could offer new advantages, as mutations in cellular genes occur infrequently, making resistance to drugs against them highly unlikely.

Apparently, PC did not influence the expression of HSV-1 proteins. The yield of icp0, icp8, and glycoprotein gD was not altered during PC administration (Appendix A). These results are in line with other studies demonstrating that siRNA-mediated knockdown of various Rab GTPases significantly reduced HSV-1 release without affecting viral protein expression of e.g., icp0, icp27, gD, gH, and gE [17,67]. Based on these results, it can be proposed that PC treatment might affect later stages of the replication cycle, such as glycoprotein processing, particle assembly, or viral release. The late stages of HSV-1 replication depend on several Rab GTPases [68]. Specifically, Rab6 facilitates the transport of viral glycoproteins through the Golgi apparatus to the plasma membrane, where they are integrated into the cell surface. Subsequently, Rab5-positive early endosomes are used by HSV-1 for transporting these glycoproteins back into the cell [26]. Depletion of Rab5 and Rab11 impairs HSV-1 replication, as shown by siRNA studies in HeLa cells [17].

Rab11 is a protein, localized in post-Golgi vesicles and recycling endosomes, that plays a key role in the transport of HSV-1 particles to the plasma membrane for the second envelopment and release [26]. The strong accumulation of these Rab5 and Rab11 vesicles in HSV-1-infected cells, in comparison to uninfected cells, illustrates the importance of both vesicles for HSV-1 replication (Figure 2a–l). PC treatment of HSV-1-infected cells prevents important cellular vesicle trafficking events, which are essential for the final virus particle assembly due to an accumulation of viral glycoprotein gD in these vesicles (Figure 2f,l and Figure 3). Therefore, the release of infectious HSV-1 is disturbed, resulting in decreased virus concentrations (Figure 1b,c).

Local anesthetics have been investigated not only for analgesia, but also for their antioxidant, anti-inflammatory, anticancer, and antiviral effects [69]. Recently, we demonstrated that PC treatment decreased the replication of SARS-CoV-2 and IAV [59,60]. Moreover, the growth of *Aspergillus fumigatus* and the viral and fungal propagation during in vitro coinfection scenarios with IAV and *Aspergillus fumigatus* were reduced as well [61]. Here, we studied whether PC exhibits antiviral potential against HSV-1 under in vitro conditions. Previously, it was reported that short-term procaine treatment in high concentrations (up to 70 mM) affected the attachment or entry of HSV-1 [54]. In this study, we did not examine the influence of these elevated concentrations on HSV-1 entry, as their extended use would likely result in cytotoxicity. Instead, our results reveal that PC in low millimolar concentrations affected a later stage of the viral replication cycle, specifically during virus assembly and release, thereby limiting the production of progeny viruses (Figure 1b,c).

HSV-1 relies on membrane reshaping to create the viral envelope. It was shown that local anesthetics affect membrane remodeling in different ways [70]. For example, PC is able to inhibit Ca^2+^-dependent phospholipase A_2_ (PLA_2_) enzymes and to block Ca^2+^ ion channels [59,60,61]. Notably, HSV-1 infection induces intracellular Ca^2+^ release that plays a key role in facilitating early events in HSV-1 invasion, intracellular trafficking, replication, and release [71,72,73]. Thus, membrane-altering properties of local anesthetics by blocking Ca^2+^ and Na^+^ ion channels might interfere with HSV-1 replication. PC is based on procaine and shows a physiological pH and increased membrane permeability [61]. Therefore, future studies might further investigate the possibility of a topical treatment for HSV-1 infections.

Two publications focusing on the HSV-1 entry show that siRNA-mediated knockdown of Rab7 in late endosomes and its overexpression did not alter viral entry processes [74,75]. In addition, pharmacological inhibition of Rab7 also had no effect on the HSV-1 replication [76]. Lysosomal vesicles are other important compartments of the cellular recycling mechanism that reprocess damaged macromolecules and target viral particles for degradation [77]. However, HSV-1 inactivates lysosomal enzymes not only to survive within lysosomes, but also exploits the lysosomal pathway to promote virus maturation and trafficking within infected cells [78,79]. Our own results indicate that PC treatment did not alter either the late endosome or the lysosomal pathway, although these are important for HSV-1 replication. Lysosomes and late endosomes were evenly distributed within uninfected and HSV-1-infected cells, without clustering with viral glycoprotein gD, as observed for Rab5 and Rab11 (Figure 2m–x, Figure 3 and Appendix A).

Virus resistance to ACV due to TK or POL polymorphisms increases in patients who require prolonged antiviral treatment and represents a major clinical challenge [45,46]. In drug-resistant strains, both clinical and in vitro studies have identified TK deficiency as the predominant phenotype, accounting for approximately 90–95% of ACV-resistant HSV isolates. In contrast, mutations in the viral DNA POL are responsible for ACV resistance in approximately 5–10% of instances [40]. In immunocompetent individuals, the prevalence of resistant strains to nucleoside analogues remains low, less than 1%. However, in immunocompromised patients infected with HSV-1, resistance rates range from 2.5% to over 30% [40]. Notably, HIV-positive individuals, organ transplant recipients, and hematopoietic stem cell transplant recipients are particularly affected by drug-resistant viruses [80,81,82]. In addition, combination therapies are commonly used during treatment of viral infections to improve efficacy, reduce toxicity, and prevent the development of drug resistance [83]. Beneficial drug combinations with synergistic or additive effects against HSV-1 infection have been identified and are established in clinical routines [44,49,51,52,84,85]. PC and ACV, both agents with antiviral potential, individually reduced viral titers, and their combined use resulted in enhanced antiviral effects (Figure 4). This suggests that the PC-ACV combination might enable lower drug concentrations without compromising antiviral efficacy.

In summary, the present study confirmed antiviral properties of PC against HSV-1 infections, together with an enhanced antiviral effect of PC and ACV on HSV-1 replication. PC has been shown to act at later stages of the HSV-1 replication cycle, possibly preventing the correct assembly of virus particles due to intracellular retention of viral proteins and/or impaired viral release. The use of PC represents a promising way to inhibit host cell-supporting functions for HSV-1 infection.

## 4. Materials and Methods

### 4.1. Cell Lines and Viruses

The hTERT RPE-1 (RPE-1, ATCC, Manassas, VA, USA), HaCaT (Cytion, Eppelheim, Germany), and Kelly cells (DSMZ, Braunschweig, Germany) were cultivated in DMEM (Anprotec, Bruckberg, Germany) supplemented with 5% fetal bovine serum (FBS, PAN^TM^Biotech, Aidenbach, Germany), except for Kelly cells, which were supplemented with 10% FBS. The HaCaT and Kelly cells were thawed at passages 58 and 24, respectively, and the experiments were conducted within a few weeks and passaging. Unless otherwise noted, all experiments were performed using the laboratory strain HSV-1 KOS (HSV-1). Some results were confirmed with the clinical HSV-1 isolate 1223/1999 of the stock of the former Institute for Virology and Antiviral Therapy Jena [86]. It was isolated from a swab of an HSV-1 infection at the eye of a patient suffering from chronic lymphocytic leukemia. Resistance to ACV was not described for this isolate. Viruses were propagated in green monkey kidney cells (Vero76, ATCC, Manassas, VA, USA) in DMEM supplemented with 10% FBS.

### 4.2. MTT Assay

To determine cell metabolic activity upon treatment with test compounds, 0.3 × 10^5^ RPE-1, 0.4 × 10^5^ HaCaT, or 0.15 × 10^5^ Kelly cells in 100 µL DMEM (supplemented with 5%, or 10% FBS, respectively) were seeded per well into 96-well plates 24 h prior to use and incubated at 37 °C and 5% CO_2_. For each experiment, substance dilutions were freshly prepared in DMEM. Cells were washed once with PBS. Then, 100 µL of DMEM with FBS, including the respective substances in specified concentrations, was added, and the cells were further incubated for 24 h at 37 °C and 5% CO_2_. Finally, 25 µL of 5 mg mL^−1^ MTT (Sigma-Aldrich^®^, Taufkirchen, Germany) was added, and cells were incubated for another 2 h. Thereafter, supernatants were carefully removed, and 30 µL of DMSO was added to lyse cells. Metabolic activity was determined via absorbance at OD 562 nm and compared to the solvent control H_2_O using a FLUOstar Omega plate reader (BMG Labtech, Ortenberg, Germany).

### 4.3. Viral Infection and TCID_50_ Titration

Per well, 0.3 × 10^6^ RPE-1, 0.5 × 10^6^ HaCaT, or 1 × 10^6^ Kelly cells were seeded in 6-well plates 24 h prior to infection. Cells were washed once with PBS and infected at the indicated MOI using DMEM without FBS for 1 h at 37 °C and 5% CO_2_. After aspiration of supernatants, cells were incubated in the presence or absence of PC (manufactured and supplied by inflamed pharma GmbH, Jena, Germany) and/or ACV, (S1807, Selleckchem, Cologne, Germany) in DMEM supplemented with 5% (RPE-1 and HaCaT) or 10% (Kelly) FBS until 24 h p.i. Supernatants were harvested to determine extracellular virus concentrations and cells were lysed for immunoblotting or fixed for immunofluorescence microscopy. To determine cell-associated virus concentrations, the infectious supernatant was replaced with fresh DMEM with 5% FBS. Subsequently, cells were subjected to three freeze-thaw cycles at −80 °C in cell culture plates. The resulting lysates from each well were transferred to a tube. After centrifugation for 5 min at 3000 rpm, supernatants were used for titration.

For TCID_50_ titration assays, 1.5 × 10^4^ Vero76 cells were seeded per well in 96-well plates 24 h prior to use. Serial dilutions of virus-containing supernatants in DMEM without FBS were added to cells. After three days of incubation at 37 °C and 5% CO_2_, cytopathic effects were visualized microscopically and values of the TCID_50_ were calculated [87].

### 4.4. Immunoblot and Antibodies

Cells were lysed with Triton lysis buffer (20 mM Tris-HCl, pH 7.4; 137 mM NaCl; 10% glycerol; 1% Triton X-100; 2 mM EDTA; 50 mM sodium glycerophosphate; 20 mM sodium pyrophosphate; 5 µg mL^−1^ aprotinin; 0.2 mM pefablock^®^; 5 µg mL^−1^ leupeptin; 1 mM sodium vanadate; and 5 mM benzamidine) for 1 h at 4 °C. After centrifugation (15 min 20,000× *g*, 4 °C), total protein content was measured (Protein Assay Dye Reagent, BioRad, Feldkirchen, Germany) and normalized using lysis buffer. Samples were supplemented with 5× Laemmli buffer (10% SDS, 50% glycerol, 25% β-mercaptoethanol, 0.02% bromophenol blue, 312 mM Tris, pH 6.8) and boiled for 10 min at 95 °C. Equal volumes were loaded on SDS-PAGE and blotted onto 0.2 µm nitrocellulose membranes. Primary antibodies were diluted 1:1000. Secondary antibodies were diluted 1:5000 (see Table 1). Membranes were developed using the ECL Western blotting substrate (Pierce^TM^, ThermoFisher Scientific, Dreieich, Germany) and the Fusion©FX6.Edge (Vilber Lourmat, Eberhardzell, Germany) with Fusion© software Evolution-Capt (Vilber Lourmat, Eberhardzell, Germany).

### 4.5. Antibody Staining for Immunofluorescence Microscopy

Per well, 0.125 × 10^6^ RPE-1 cells were seeded in 24-well plates supplemented with sterile glass coverslips (CarlRoth^®^, Karlsruhe, Germany, #1; 0.13–0.16 mm thickness) 24 h prior to infection, and infected and treated with the substances as described above.

After 24 h, cells were washed once with PBS, fixed with 3.7% formaldehyde/PBS (Sigma-Aldrich^®^, Taufkirchen, Germany) for 15 min, washed three times with PBS, permeabilized with 0.1% Triton-X100/PBS (Sigma-Aldrich^®^, Taufkirchen, Germany) for 15 min, washed three times with PBS, and blocked with 3% BSA/PBS (CarlRoth^®^, Karlsruhe, Germany) for 30 min. Primary antibodies were diluted in 3% BSA/PBS and incubated overnight at 4 °C. After washing three times with PBS, secondary antibodies and dyes were also diluted in 3% BSA/PBS and incubated for 1 h at room temperature. Hoechst 33342 (Sigma-Aldrich^®^, Taufkirchen, Germany) was used to visualize cellular DNA, and Phalloidin-iFluor™ 647 (Biomol, Hamburg, Germany) was used to stain F-actin (see Table 2). After final three washing steps with PBS, one drop of non-hardening mounting medium (Ibidi, Gräfelfing, Germany) was added. All slides were stored at 4 °C.

### 4.6. Wide-Field Fluorescence Microscopy

The 4-channel z-stacks were acquired as 14-bit images using an AXIO Observer.Z1 microscope (Zeiss, Jena, Germany) equipped with ApoTome.2 (Zeiss, Jena, Germany), Axiocam 503 mono (Zeiss, Jena, Germany), an HXP 120 V light source (Leistungselektronik Jena GmbH, Jena, Germany), and Zen blue version 2.6 (Zeiss, Jena, Germany). For image acquisition, the following filter sets (Zeiss, Jena, Germany) were used for the respective dyes: 50 (excitation (ex): 625–655 nm, emission (em): 665–715 nm) for AF647, 43 (ex: 533–558 nm, em: 570–640 nm) for Cy3, 38HE (ex: 450–490 nm, em: 500–550 nm) for AF488, and 49 (ex: 335–383 nm, em: 420–470 nm) for DAPI. Images were acquired using a Plan-Apochromat 20×/0.8 NA air objective (Zeiss, Jena, Germany) and a fixed voxel size of 0.227 µm in xy and 0.490 µm z, or an LD Plan-Neofluar 40×/0.6 NA air objective (Zeiss, Jena, Germany) and a fixed voxel size of 0.114 µm in xy and 0.490 µm z, with light intensities and exposure times as indicated in Table 3. The resulting approximate resolution for the two channels subjected to co-localization analysis was calculated according to the Abbe formulation (d = λ/(2∗NA)) as follows: 455 nm for the Cy3 channel (gD signal) and 392 nm for the AF488 channel (endosomal or lysosomal marker proteins). After acquisition, images were directly deconvolved in Zen blue with phase error correction and a deconvolution strength of 6. Deconvolved z-stacks were used for further processing, co-localization analysis, and quantification in Fiji Is Just ImageJ (FIJI V. 2.16.0/1.54p) [88]. For image display, deconvolved z-stacks were maximum intensity projected in Zen blue and the dynamic ranges were adjusted as described in Table 3.

### 4.7. Co-Localization Analysis

The 4-channel z-stacks (red channel: actin; orange channel: gD; green channel: endosomal or lysosomal marker; blue channel: nuclei) were loaded into FIJI (V. 2.16.0/1.54p) as 16-bit images using the Bio-Formats Importer (V. 8.0.0) [88,89]. In a first step, channels were split and only the blue channel (nuclei) was processed to receive regions of interest (ROIs) based on Voronoi diagrams to approximate cell shapes. Blue-channel z-stacks were maximum intensity projected and the resulting maximum intensity projections (MIPs) were processed with the following steps: rolling ball background subtraction (radius = 500), ball-shaped median filter (radius = 5), and Gaussian Blur filter (sigma = 15). Images were then thresholded using the Triangle method [90]. Binary images were subjected to Adjustable Watershed separation (tolerance = 1) and three Erode iterations. Finally, the Voronoi filter was applied, the resulting mask was thresholded with upper and lower threshold values of 0, and the resulting ROIs were added to the ROI manager and extracted as zip files.

Co-localization analysis was performed on individual z-slices. In order to determine which z-slice had to be used, the orange (gD) and green (endosomal or lysosomal marker) z-stacks were screened individually for the slice with the highest mean intensity (‘brightest slice’). Since the brightest slice always differed slightly between both channels, the central slice between them was subjected to co-locali zation analyses. In the case of an odd difference between both brightest slices, the lower z-plane was chosen (examples shown in Table 4). Then, Pearson’s and Manders’ thresholded coefficients were determined per ROI using the JACoP BIOP plugin (V. 1.2.0) [91,92,93,94,95]. Within the plugin, the orange channel (gD) was thresholded using the Li method, while the green channel (endosomal or lysosomal marker) was thresholded using the Triangle method for the determination of the Manders’ thresholded coefficients [90,96]. In total, three images per condition and experiment were analyzed, and the median Pearson’s/Manders’ coefficient of all ROIs from three images was calculated. Once data from several experiments were obtained, these median values were plotted, and means ± standard deviation were calculated.

### 4.8. Quantification of the gD Fluorescence Signal

The gD fluorescence signals were quantified based on MIPs. Therefore, 4-channel z-stacks were loaded into FIJI (V. 2.16.0/1.54p) as 16-bit images using the Bio-Formats Importer (V. 8.0.0) [88,89]. Channels were split and only the orange (gD) channel was further processed. Z-stacks were maximum intensity projected and mean fluorescence intensities per image were determined using FIJI’s inbuilt measure function. In total, three experiments with four technical replicates and three images per replicate were analysed (resulting in 12 images per experiment). The median mean fluorescence intensity per image was determined for each technical replicate and data from technical replicates were combined into one mean value for each experiment. Once data from several experiments were obtained, these mean values were plotted and means (±SD) were calculated.

### 4.9. Statistics

Statistical analyses were performed using GraphPad Prism 8 software and are described in the respective figure legends.

## 5. Patents

B.E. has patent #WO 2021/198504 A1 issued to inflamed pharma GmbH and patent #WO 2019/048590 A1 issued to JenCluster GmbH.

## Figures and Tables

**Figure 1 ijms-26-07185-f001:**
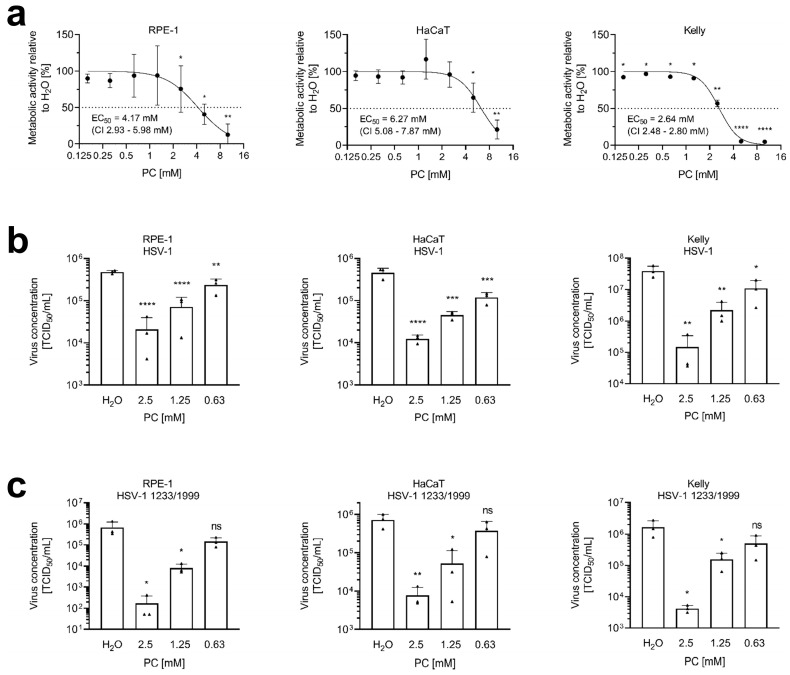
ProcCluster^®^ (PC) inhibits HSV-1 replication at non-toxic concentrations. (**a**) RPE-1, HaCaT, and Kelly cells were cultured in the presence or absence of PC for 24 h. Metabolic activity was determined using MTT assays. Results were normalized to H_2_O-treated control cells, which were arbitrarily set to 100%. Data are presented as the mean ± SD of three independent experiments including three biological replicates. Statistical significance relative to the 100% reference was analyzed by a two-tailed one-sample *t*-test (* *p* < 0.05; ** *p* < 0.01; **** *p* < 0.0001). Effective concentration 50% (EC_50_) values and 95% confidence intervals (CI) were determined by non-linear regression using GraphPad Prism software. (**b**,**c**) RPE-1, HaCaT, or Kelly cells were infected with either (**b**) the laboratory strain HSV-1 or (**c**) the clinical isolate HSV-1 1223/1999, both at an MOI of 1.0 for 1 h. Cells were further incubated in the presence or absence of PC. At 24 h post-infection (p.i.), supernatants were collected and progeny virus titers were determined by tissue culture infectious dose 50% (TCID_50_) titrations. The mean + SD of three independent experiments including two biological replicates, is depicted. Statistical significance compared to H_2_O was analyzed by ordinary one-way ANOVA followed by Dunnett’s multiple comparisons test (ns, not significant; * *p* < 0.05; ** *p* < 0.01; *** *p* < 0.001; **** *p* < 0.0001).

**Figure 2 ijms-26-07185-f002:**
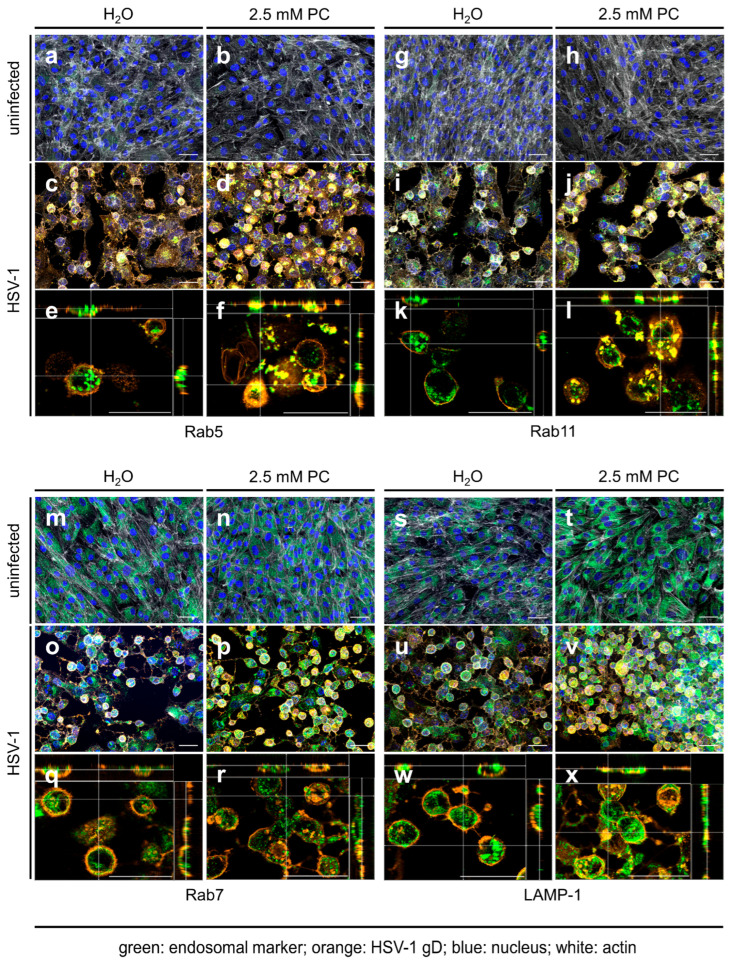
HSV-1 glycoprotein gD accumulates in early and recycling endosomes but not in late endosomes and lysosomes after PC treatment. RPE-1 cells were infected with HSV-1 using an MOI of 1.0 for 1 h or were left uninfected. Cells were further incubated in the presence or absence of 2.5 mM PC. At 24 h p.i., cells were fixed, permeabilized, and stained for HSV-1 gD (orange), endosomal markers (green), F-actin (white), and nuclei (blue). (**a**–**d**,**g**–**j**,**m**–**p**,**s**–**v**) Z-stacks were acquired using a 20× objective. Images were deconvolved, and maximum intensity projections with all 4 channels merged are shown. (**e**,**f**,**k**,**l**,**q**,**r**,**w**,**x**) Z-stacks were acquired using a 40× objective and deconvolved as above. Shown are orthogonal views with only the orange and green channels merged. The central xy image shows a single z-slice; side views show the xz and yz planes at the positions indicated by white crosshairs. (**a**–**x**) Scale bars represent 50 µm. Representative pictures of three independent experiments are displayed.

**Figure 3 ijms-26-07185-f003:**
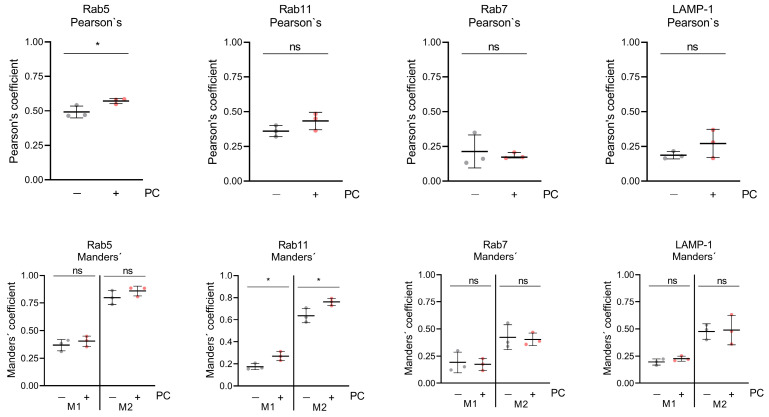
PC treatment increases the correlation and co-occurrence coefficients of Rab5 as well as Rab11, but not those of Rab7 or LAMP1, with HSV-1 glycoprotein gD. The blue channel (nuclei) was processed to receive regions of interest (ROIs) based on Voronoi diagrams in order to approximate cell shapes. Then, Pearson’s and Manders’ thresholded coefficients were determined per ROI. For Manders’, both thresholded coefficients, M1, the fraction of gD signal that co-occurs with protein of interest (POI) signal, and M2, the fraction of POI signal that co-occurs with gD signal, are shown. In total, three images per condition and experiment were analyzed and depicted as the mean (±SD). Statistical significance was analyzed by unpaired two-tailed *t*-test (* *p* < 0.05; ns, not significant).

**Figure 4 ijms-26-07185-f004:**
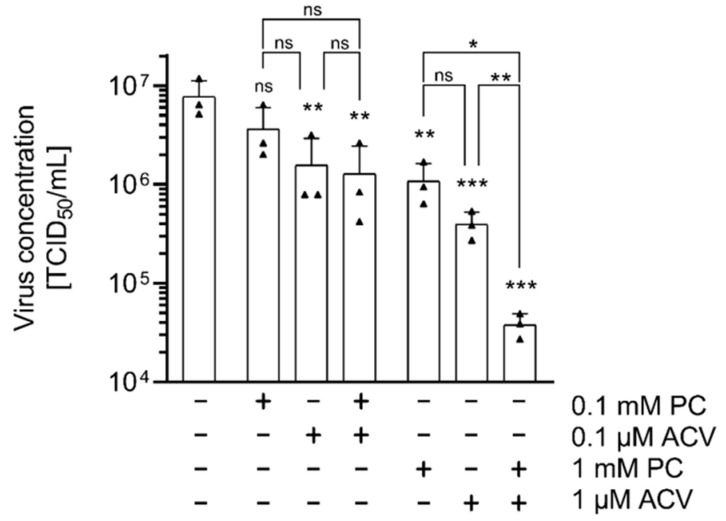
The combined use of PC and ACV shows enhanced antiviral effects on HSV-1 replication in vitro. RPE-1 cells were infected with HSV-1 at an MOI of 1.0 for 1 h and subsequently incubated in the presence or absence of different PC and ACV concentrations. At 24 h p.i., supernatants were collected and progeny virus titers were determined by TCID_50_ titrations. The mean of three independent experiments, each including two biological replicates, is depicted. Statistical significance compared to H_2_O was analyzed by ordinary one-way ANOVA followed by Dunnett’s multiple comparisons test, and between the combinations by unpaired two-tailed *t*-test (ns, not significant; * *p* < 0.05; ** *p* < 0.01; *** *p* < 0.001).

**Table 1 ijms-26-07185-t001:** Antibodies for immunoblotting.

Antibody	Source and Catalogue Number
mAb HSV-1 icp0	sc-53070 (Santa Cruz Biotechnology, Heidelberg, Germany)
mAb HSV-1 icp8	sc-53329 (Santa Cruz Biotechnology, Heidelberg, Germany)
mAb HSV-1 gD	sc-21719 (Santa Cruz Biotechnology, Heidelberg, Germany)
mAb α-tubulin	2125 (Cell signaling Technology, Wetzlar, Germany)
pAb HSP90	4877 (Cell signaling Technology, Wetzlar, Germany)
WesternSure^®^ HRP goat anti-mouse IgG	926-80010 (LICORbio, Bad Homburg, Germany)
WesternSure^®^ HRP goat anti-rabbit IgG	926-80011 (LICORbio, Bad Homburg, Germany)

**Table 2 ijms-26-07185-t002:** Antibodies and dyes for immunofluorescence microscopy.

Antibody	Source and Catalogue Number
mAb HSV-1 gD (1:125)	sc-21719 (Santa Cruz Biotechnology, Heidelberg, Germany)
mAb Rab11 (1:125)	5589 (Cell signaling Technology, Wetzlar, Germany)
mAb Rab5 (1:125)	3547 (Cell signaling Technology, Wetzlar, Germany)
mAb Rab7 (1:125)	9367 (Cell signaling Technology, Wetzlar, Germany)
mAb LAMP1 (1:125)	9091 (Cell signaling Technology, Wetzlar, Germany)
AffiniPure Goat Anti-Mouse Cy 3 (1:500)	115-165-003 (Dianova, Hamburg, Germany)
AffiniPure Goat Anti-Rabbit IgG Alexa Fluor 488 (1:500)	111-545-144 (Dianova, Hamburg, Germany)
Hoechst 33342 (1:1000)	14533 (Sigma-Aldrich^®^, Taufkirchen, Germany)
Phalloidin-iFluor^TM^ 647 (1:1000)	ABD-23127 (Biomol, Hamburg, Germany)

**Table 3 ijms-26-07185-t003:** Image acquisition parameters and dynamic ranges displayed in Figure 2.

Figure	Component	Chromophore	Light Intensity [%]	Exposure Time [ms]	Displayed Dynamic Range
a–d	Actin	iFluor^TM^ 647	100	1000	200–10,000
	HSV-1 gD	Cy3	70	70	200–4000
	Rab5	Alexa Fluor 488	80	400	200–4000
	Nucleus	Hoechst33342	80	200	200–7000
e,f	HSV-1 gD	Cy3	80	100	200–3000
	Rab5	Alexa Fluor 488	90	650	200–2000
g–j	Actin	iFluor^TM^ 647	100	600	200–8000
	HSV-1 gD	Cy3	70	50	200–4000
	Rab11	Alexa Fluor 488	100	1000	200–4000
	Nucleus	Hoechst33342	80	100	200–7000
k,l	Nucleus	Hoechst33342	80	100	200–7000
	HSV-1 gD	Cy3	100	400	30–2000
m–p	Actin	iFluor^TM^ 647	100	800	200–6000
	HSV-1 gD	Cy3	70	50	200–3000
	Rab7	Alexa Fluor 488	100	1000	200–3000
	Nucleus	Hoechst33342	80	150	200–5000
q,r	HSV-1 gD	Cy3	100	400	200–4000
	Rab7	Alexa Fluor 488	100	5000	200–4000
s–v	Actin	iFluor^TM^ 647	100	500	200–7000
	HSV-1 gD	Cy3	80	90	200–5000
	LAMP1	Alexa Fluor 488	100	400	200–5000
	Nucleus	Hoechst33342	80	70	200–7000
w,x	HSV-1 gD	Cy3	100	200	200–3000
	LAMP1	Alexa Fluor 488	100	2500	200–2000

**Table 4 ijms-26-07185-t004:** Examples for the determination of which z-slice was subjected to co-localization analysis.

Brightest Slice in Orange Channel	Brightest Slice in Green Channel	Chosen Slice for Co-Localization Analysis
19	21	20
23	27	25
20	23	21
17	22	19

## Data Availability

The data presented in this study are available in the present article and Appendix A.

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
