# Peer review of "The Procaine-Based ProcCluster® Impedes the Second Envelopment Process of Herpes Simplex Virus Type 1"

_ijms, 2025, doi:10.3390/ijms26157185_

Round 1

Reviewer 1 Report

Comments and Suggestions for Authors

The present study investigated anti-viral effects of procaine based cluster (PC) on two HSV-1 isolates, alone and in combination with acyclovir. Overall the study is well performed and the manuscript is well written.

  1. For figures 2-5, I recommend graphing colocalization coefficients for certain number of infected cells for a better interpretation. Its hard to interpret the images the authors have shown in the results. In addition, I d also recommend included the total gD intensity as a measure of total infection between H2o and PC-treated cells
  2. Discussion section should include possible PC-targeted pathways that could ve prevented egress.

Author Response

Reviewer 1

1: For figures 2-5, I recommend graphing colocalization coefficients for certain number of infected cells for a better interpretation. Its hard to interpret the images the authors have shown in the results. In addition, I d also recommend included the total gD intensity as a measure of total infection between H2o and PC-treated cells.

Response: We followed the suggestion of the reviewer. We analyzed the colocalization and gD intensity. Please note, that we changed the arrangement of the figures due to a comment of the second reviewer. Old figures 2-5 are now shown in Figure 2.

The results of the co-localization coefficients are depicted in the new Figure 3, with individual experiments in the new supplementary figure S2 and the results of the gD quantification are depicted in new supplementary figure S1 c.

Due to the strong HSV-1-induced cytopathic effect it was difficult to segment the cells. We finally segmented the images based on a Voronoi mask received from nuclei channel. In this case, some images contain too many or too small regions of interest (ROIs), due to cells overlapping or weak DAPI signal, which might influence the co-localization results.

From all the co-localization coefficients we choose to analyse the Person`s and Manders´ thresholded coefficients, which itself have certain advantages and disadvantages. Although Pearson`s is independent on thresholding, it is highly dependent on ROIs. Manders` on the other hand is highly dependent on thresholding (introducing bias by subjective choosing the thresholding method) but less dependent on black pixels/ROI positioning. For this reason, we used the more specific terms correlation and co-occurence instead of colocalization.

We clarified the subsection microscopy: (line 436 – 450)

For image acquisition, the following filter sets (Zeiss, Germany) were used for the respective dyes: 50 (excitation (ex): 625-655 nm, emission (em): 665-715 nm) for AF647, 43 (ex: 533-558 nm, em: 570-640 nm) for Cy3, 38HE (ex: 450-490 nm, em: 500-550 nm) for AF488, and 49 (ex: 335-383 nm, em: 420-470 nm) for DAPI. Images were acquired using a Plan-Apochromat 20x/0.8 NA air objective (Zeiss, Germany) and a fixed voxel size of 0.227 µm in xy and 0.490 µm z, or a LD Plan-Neofluar 40x/0.6 NA air objective (Zeiss, Germany) and a fixed voxel size of 0.114 µm in xy and 0.490 µm z, with light intensities and exposure times as indicated in Table 3. The resulting approximate resolution for the two channels subjected to co-localization analysis was calculated according to the Abbe formulation (d = λ/(2*NA)) as follows: 455 nm for the Cy3 channel (gD signal) and 392 nm for the AF488 channel (endosomal or lysosomal marker proteins). After acquisition, images were directly deconvolved in Zen blue with phase errors correction and a deconvolution strength of 6. Deconvolved z-stacks were used for further processing, co-localization analysis, and quantification in Fiji Is Just ImageJ (FIJI V. 2.16.0/1.54p) [88].

Furthermore, we included two new subsections in the methods section: (line 454 – 494)

4.7. Co-localization analysis

Four channel z-stacks (red channel: actin; orange channel: gD, green channel: endosomal or lysosomal marker, blue channel: nuclei) were loaded into FIJI (V. 2.16.0/1.54p) as 16-bit images using the Bio-Formats Importer [88,89]. In a first step, channels were split and only the blue channel (nuclei) was processed to receive regions of interest (ROIs) based on Voronoi diagrams to approximate cell shapes. Blue-channel z-stacks were maximum intenstity projected and the resulting maximum intenstity projections (MIPs) were processed with the following steps: rolling ball background subtraction (radius = 500), ball-shaped median filter (radius = 5), and Gaussian Blur filter (sigma = 15). Images were then thresholded using the Triangle method [90]. Binary images were subjected to Adjustable Watershed separation (tolerance = 1) and three Erode iterations. Finally, the Voronoi filter was applied, the resulting mask was thresholded with upper and lower threshold values of 0, and the resulting ROIs were added to the ROI manager and extracted as zip files.

Co-localization analysis was performed on individual z-slices. In order to determine which z-slice had to be used, the orange (gD) and green (endosomal or lysosomal marker) z-stacks were screened individually for the slice with the highest mean intensity (‘brightest slice’). Since the brightest slice always differed slightly between both channels, the central slice between them was subjected to co-localization analyses. In the case of an odd difference between both brightest slices, the lower z-plane was chosen (examples shown in Table 4). Then, Pearson’s and Manders’ thresholded coefficients were determined per ROI using the JACoP BIOP plugin [91-95]. Within the plugin, the orange channel (gD) was thresholded using the Li method, while the green channel (endosomal or lysosomal marker) was thresholded using the Triangle method for the determination of the Manders’ thresholded coefficients [90,96]. In total, three images per condition and experiment were analysed and the median Pearson’s/Manders’ coefficient of all ROIs from three images was calculated. Once data from several experiments were obtained, these median values were plotted and means ± standard deviation were calculated.

Table 4. Examples for the determination of which z-slice was subjected to co-localization analysis. (table, please see manuscript)

4.8. Quantification of the gD fluorencence signal

gD fluorescence signals were quantified based on MIPs. Therefore, 4 channel z-stacks were loaded into FIJI (V. 2.16.0/1.54p) as 16-bit images using the Bio-Formats Importer [88,89]. Channels were split and only the orange (gD) channel was further processed. Z-stacks were maximum intensity projected and mean fluorescence intensities per image were determined using FIJI’s inbuilt measure function. In total, three experiments with four technical replicates and three images per replicate were analysed (resulting in 12 images per experiment). The median mean fluorescence intensity per image was determined for each technical replicate and data from technical replicates were combined to one mean value for each experiment. Once data from several experiments were obtained, these mean values were plotted and means (± SD) were calculated.

We further included the following informations in the results section:

(line 199 – 214): Consistent with the cell-associated total viral concentration (Figure S1 b), measurement of the mean fluorescence intensity of viral glycoprotein gD in PC-treated, HSV-1-infected RPE-1 cells revealed no in-creased intensity (Figure S1 c).

(line 220 – 224): PC treatment, however, resulted in accumulation of larger intracellular vesicles and a strong correlation and co-occurrence of viral glycoprotein gD together with Rab5/Rab11, respectively, within these vesicles (Figures 2 d, f and 2 j, l, 3, and S2). Both, the Pearson`s correlation, as well as the Manders´ thresholded coefficients show a significant increase in their respective coefficients (Figure 3).

(line 234 – 238): To analyze, whether that correlation and co-occurrence of viral glycoprotein gD with Rab5/Rab11 in the presence of PC is restricted to early and recycling endosomes, respectively, the viral glycoprotein gD was additionally assayed for PC-induced co-localization with the late endosomal marker protein Rab7 (Figures 2 m-r) and the lysosomal marker protein LAMP-1 (Figures 2 s-x).

(line 240 – 245): Strikingly, PC treatment resulted again in enlarged viral glycoprotein gD clusters but this time neither an increased correlation, nor an increased co-occurrence was observed with either Rab7 or LAMP-1 (Figures 2 q, r and 2 w, x, 3, and S2). In parallel, both Pearson`s coefficients for Rab7 and LAMP-1 are lower, demonstrating itself a weaker correlation between gD and both markers.

We further included the figure legend S1: (line 515 – 521)

(c) The gD fluorescence signals were quantified based on MIPs. Z-stacks were maximum intensity projected and mean fluorescence intensities per image were determined. Three experiments with four technical replicates and three images per replicate were analyzed using FIJI’s inbuilt measure function. The median mean fluorescence intensity per image was determined for each technical replicate, combined to one mean value for each experiment and depicted as means (± SD). Statistical significance was analyzed by unpaired two-tailed t-test (** p<0.001; ns, not significant).

We further included the figure legend S2: (line 522 – 528):

PC treatment increases correlation and co-occurrence coefficients of Rab5 as well as Rab11, but not those of Rab7 or LAMP1, with HSV-1 glycoprotein gD. The blue channel (nuclei) was pro-cessed to receive ROIs based on Voronoi diagrams to approximate cell shapes. Then, Pearson’s and Manders’ thresholded coefficients were determined per ROI. For Manders´, both thresholded coefficients, M1, the fraction of gD signal that co-occurs with POI signal, and M2, the fraction of POI signal that co-occurs with gD signal, are shown. In total, three images per condition and experiment (exp) were analyzed and depicted as the median with interquartile range.

2: Discussion section should include possible PC-targeted pathways that could ve prevented egress.

Response: Since local anesthetics are interfering with intracellular ion channels, we clarified our discussion with the following: (line 312 – 321)

HSV-1 relies on membrane reshaping to create the viral envelope. It was shown that local anesthetics affects membrane remodeling in different ways [70]. For example, PC is able to inhibit Ca2+-dependent phospholipase A2 (PLA2) enzymes and to block Ca2+ ion channels [59-61]. Notably, HSV-1 infection induces intracellular Ca2+ release that plays a key role in facilitating early events in HSV-1 invasion, intracellular trafficking, replication, and also release [71-73]. Thus, membrane altering properties of local anesthetics by blocking Ca2+ and Na+ ion channels might interfere with HSV-1 replication. PC is based on procaine and shows improved water solubility, a physiological pH, and increased mem-brane permeability [61]. Therefore, future studies might further investigate the possibility of a topical treatment for HSV-1 infections.

Reviewer 2 Report

Comments and Suggestions for Authors

The manuscript by Jungwirth et al describes antiviral effect of the local anasthetic Procaine, specifically against HSV-1. As this is a pathogen of the mucus membranes the relevance of Procaine is obvious. The paper describes a novel antiviral effect, and although this agent is known to act against several viruses this is a relatively novel application. More discussion of the clinical relevance of the finding would be good, and more discussion on the possible mechanism of inhibition of gD in the VAC would also be important.

Figures should be reformatted for clarity. In particular figures 2-5 should be combined into a single large figure. As this figure is primarily cellular localisation of the viral gD protein flow cytometry of the gD protein on the cell surface should be shown. in combination with the western blot this would be important for interpretation of the result

Minor points. 
HaCaT cell passage number should be noted, as this cell line changes upon prolonged culture and becomes more mature.
The origin of the clinical HSV-1 isolate should be noted, and if it is resistant.
Antibody type should be noted (mAb or pAb) as this is important for interpretation of the pictures.

Comments on the Quality of English Language

OK, The english used is in places a little stiff, and could do with rewording

Author Response

Reviewer 2

1: More discussion of the clinical relevance of the finding would be good and more discussion on the possible mechanism of inhibition of gD in the VAC would also be important.

Response: We described the relevance of resistance against direct acting antivirals and of ACV-resistant strains in particular in immunocompromised patients (line 334 – 351). PC appears to target virus-supporting cellular factors, which could be beneficial. However, future studies need to investigate this in more detail.

Since local anesthetics are interfering with intracellular ion channels we clarified our discussion and included the following: (line 312 – 321)

HSV-1 relies on membrane reshaping to create the viral envelope. It was shown that local anesthetics affects membrane remodeling in different ways [70]. For example, PC is able to inhibit Ca2+-dependent phospholipase A2 (PLA2) enzymes and to block Ca2+ ion channels [59-61]. Notably, HSV-1 infection induces intracellular Ca2+ release that plays a key role in facilitating early events in HSV-1 invasion, intracellular trafficking, replication, and also release [71-73]. Thus, membrane altering properties of local anesthetics by blocking Ca2+ and Na+ ion channels might interfere with HSV-1 replication. PC is based on procaine and shows improved water solubility, a physiological pH, and increased mem-brane permeability [61]. Therefore, future studies might further investigate the possibility of a topical treatment for HSV-1 infections.

2: Figures should be reformatted for clarity. In particular figures 2-5 should be combined into a single large figure.

Response: As suggested, we rearranged the figures 2-5 into one large figure 2. Concomitantly, we also had to change the figure legend, the links within the text, and the table 3. Please note, that we also quantified the co-localization.

3: As this figure is primarily cellular localisation of the viral gD protein flow cytometry of the gD protein on the cell surface should be shown. in combination with the western blot this would be important for interpretation of the result

Response: You are correct. We can`t show HSV-1 proteins on the cell surface because HSV-1 virions are released of the cell via exocytosis. But nevertheless, we performed additional experiments to examine cell-associated HSV-1 concentrations that aren`t released of the cells and we quantified the viral gD intensity of the immunofluorescence images. The results are shown in the new figures S1 b and c.

We added the following information in the methods section: (line 392 – 396)

To determine cell-associated virus concentrations, the infectious supernatant was replaced with fresh DMEM with 5% FBS. Subsequently, cells were subjected to three freeze-thaw cycles at -80°C in cell culture plates. The resulting lysates from each well were transferred to a tube. After centrifugation for 5 min at 3,000 rpm, supernatants were used for titration.

(line 484 – 494)

The gD fluorescence signals were quantified based on MIPs. Therefore, 4 channel z-stacks were loaded into FIJI (V. 2.16.0/1.54p) as 16-bit images using the Bio-Formats Importer [88,89]. Channels were split and only the orange (gD) channel was further processed. Z-stacks were maximum intensity projected and mean fluorescence intensities per image were determined using FIJI’s inbuilt measure function. In total, three experiments with four technical replicates and three images per replicate were analysed (resulting in 12 images per experiment). The median mean fluorescence intensity per image was determined for each technical replicate and data from technical replicates were combined to one mean value for each experiment. Once data from several experiments were obtained, these mean values were plotted and means (± SD) were calculated.

We added the following information in the results section: (line 173 – 192)

This was confirmed by comparison of released and cell associated viral particles. Analysis of the concentration of cell-associated viral particles at 24 h p.i. in PC-treated, HSV-1-infected RPE-1 cells revealed no reduction in infectious virions compared to the untreated controls (Figure S1 b).

 (line 199 – 214): Consistent with the cell-associated total viral concentration (Figure S1 b), measurement of the mean fluorescence intensity of viral glycoprotein gD in PC-treated, HSV-1-infected RPE-1 cells revealed no in-creased intensity (Figure S1 c).

We added the following information in the figure legend of S1: (line 503 – 521)

(b) RPE-1 cells were infected with HSV-1 at an MOI of 1.0 for 1 h and were further incubated in the presence and absence of PC. At 24 h p.i., supernatants were collected and progeny virus titers were determined by TCID50 titrations. Cell-associated virus titers were determined in fresh DMEM after 3 freeze-thaw cycles at -80°C. The mean + SD of three independent experiments including two biological replicates is depicted. Statistical significance compared to solvent control was analyzed by unpaired two-tailed t-test (* p=0.0464; ns not significant). (c) The gD fluorescence signals were quantified based on MIPs. Z-stacks were maximum intensity projected and mean fluorescence intensities per image were determined. Three experiments with four technical replicates and three images per replicate were analyzed using FIJI’s inbuilt measure function. The median mean fluorescence intensity per image was determined for each technical replicate, combined to one mean value for each experiment and depicted as means (± SD). Statistical significance was analyzed by unpaired two-tailed t-test (** p<0.001; ns, not significant).

4: HaCaT cell passage number should be noted, as this cell line changes upon prolonged culture and becomes more mature.

Response: You are correct, we added the following information in the cell lines and viruses subsection: (line 363 – 365)

The HaCaT and Kelly cells were thawed at passage 58 and 24, respectively, and the experiments were done within a few weeks and passaging.

5: The origin of the clinical HSV-1 isolate should be noted, and if it is resistant.

Response: At first, please let us correct the ID of the isolate to 1223/1999. We added a published reference of us and the following in the cell lines and viruses subsection: (line 367 – 369)

Some results were confirmed with the clinical HSV-1 isolate 1223/1999 of the stock of the former Institute for Virology and Antiviral Therapy Jena [86]. It was isolated from a swab of an HSV-1 infection at the eye of a patient suffering from chronic lymphocytic leukemia. Resistance to ACV was not described for this isolate.

6: Antibody type should be noted (mAb or pAb) as this is important for interpretation of the pictures.

Response: That`s correct. We added the according information in table 1 and 2, respectively.

Round 2

Reviewer 2 Report

Comments and Suggestions for Authors

RE:  response to point 3:

"We can`t show HSV-1 proteins on the cell surface because HSV-1 virions are released of the cell via exocytosis."

Oops - not so. In vitro syncitial formation can happen for all strains of HSV1 and 2 (not just HSV strain 17) as viral entry proteins appear at the cell surface quite soon after infection; they are not only associated with the virion. As this paper is concerned with trafficing of viral proteins/formation of the VAC this is relevant.